# The Influence of Servant Leadership on the Professional Well-Being of Kindergarten Teachers: A Moderated Mediation Model

**DOI:** 10.3390/bs15101412

**Published:** 2025-10-17

**Authors:** Dongqing Yu, Wenxin Yue, Shuang Hao, Dengyin Li, Qiurong Wu

**Affiliations:** 1Faculty of Education, Northeast Normal University, Changchun 130024, China; yudq049@nenu.edu.cn (D.Y.); yuewenxin@nenu.edu.cn (W.Y.); wuqr153@nenu.edu.cn (Q.W.); 2Faculty of Humanities, Tianjin University of Finance and Economics Pearl River College, Tianjin 301811, China; 3Learning Sciences and Technologies (STEM), University of Pennsylvania, Philadelphia, PA 19104, USA; nie25.ld1623@e.ntu.edu.sg

**Keywords:** servant leadership, professional well-being, psychological empowerment, inclusive climate, kindergarten teachers, moderated mediation model

## Abstract

This study examined the relationships among principals’ servant leadership, kindergarten teachers’ professional well-being, psychological empowerment, and an inclusive atmosphere. A questionnaire survey was conducted with 531 kindergarten teachers using purposive sampling. Results showed that (1) principals’ servant leadership positively predicted teachers’ professional well-being; (2) servant leadership positively predicted teachers’ psychological empowerment; (3) psychological empowerment mediated the relationship between servant leadership and teachers’ professional well-being; and (4) an inclusive kindergarten climate moderated the relationship between servant leadership and psychological empowerment. These findings clarify the mechanism through which servant leadership influences teachers’ professional well-being and provide practical implications for improving kindergarten management and promoting teachers’ occupational well-being.

## 1. Introduction

Teacher occupational well-being (TWB) is a critical issue for both society and schools ([24]), as it plays an essential role in enhancing teachers’ professional competence and supporting learner development ([43]; [37]; [77]). TWB encompasses teachers’ sense of achievement, self-efficacy, and overall job satisfaction. It is not only a positive emotional state but also a key psychological indicator of work quality ([6]; [1]), especially under high job demand and work pressure TWB also influences the quality of teacher–child interactions ([49]) as well as children’s growth, development, and personality formation ([37]; [26]). In recent years, rising turnover among kindergarten teachers has attracted growing attention across fields ([9]; [73]). Identifying the factors that shape their occupational well-being is therefore essential for stabilizing the workforce and improving the overall quality of preschool education.

Previous research indicates that both personal and organizational characteristics influence teachers’ professional well-being. On the personal level, psychological factors such as psychological well-being ([55]), cognitive regulation ([33]), psychological capital ([78]; [43]), and psychological empowerment ([57]) are significant determinants. At the organizational level, beyond support ([22]), job stress ([31]), and organizational climate ([28]), leadership style has been identified as one of the most critical predictors of teachers’ professional well-being ([51]; [12]; [38]). Servant leadership, in particular, is a teacher-centered approach rooted in service-oriented values. It emphasizes prioritizing teachers’ interests, addressing their needs, fostering professional growth through empowerment, and creating a supportive psychological climate. Compared with other leadership styles, servant leadership is especially attentive to teachers’ professional needs and long-term development ([23]). Taken together, teachers’ professional well-being is shaped by the interplay of internal psychological resources and external organizational conditions.

Since the 1930s, “happiness” has gained prominence as an academic focus ([42]). In recent years, educators’ professional well-being has also attracted growing societal concern ([20]). In China, however, the preschool teaching workforce continues to face challenges such as heavy workloads, relatively low overall qualifications, high occupational mobility, and elevated turnover rates ([56]). Addressing kindergarten teachers’ professional well-being and identifying effective strategies to enhance it have thus become pressing issues. This study aims to improve the research on kindergarten teachers’ professional well-being to investigate how the servant leadership by kindergarten principals affects the professional well-being of kindergarten teachers and to explore the roles of psychological empowerment and an inclusive atmosphere.

## 2. Literature Review

### 2.1. Servant Leadership and Teachers’ Professional Well-Being

Previous research has shown that leaders’ behaviors, leader-employee relationships, and leadership styles are significantly related to employees’ stress levels and affective well-being ([61]). Servant leadership differs from transformational leadership in both vision formation and trust-building mechanisms, as it emphasizes member well-being ([68]) and cultivates trust primarily through emotional bonds rather than cognitive strategies or organizational interests ([59]). As a result, this leadership style is more effective in eliciting employees’ emotional identification, thereby enhancing their professional well-being.

From a theoretical perspective, servant leadership promotes human values and community-oriented practices by addressing employees’ developmental and psychological needs ([19]; [23]). Leaders who serve as role models and communicate effectively can strengthen employees’ professional well-being through supportive human resource management ([21]; [2]). The Job Demands-Resources (JD-R) model further explains this process, suggesting that job resources provided by leaders interact with employees’ psychological empowerment to mitigate the negative impact of job demands and foster occupational well-being ([4]). Empirical studies have confirmed that servant leadership is positively related to teachers’ well-being ([29]; [12]) and can reduce stress, alleviate burnout, and enhance professional happiness by providing sufficient resources ([14]; [65]).

Despite extensive evidence from fields such as business and healthcare, empirical research on the impact of servant leadership on well-being in education, particularly in early childhood education, remains limited. This gap highlights the necessity for further investigation into servant leadership as a pathway to enhancing kindergarten teachers’ professional well-being. Thus, the first hypothesis of this study is:

**H1.** 
*Servant leadership by principals significantly predicts kindergarten teachers’ professional well-being.*


### 2.2. Psychological Empowerment as a Mediator

Psychological empowerment was first introduced by [11] ([11]) and further developed by [62] ([62]) as a psychological state shaped by individuals’ holistic perceptions of their work environment. It encompasses four dimensions: job meaning, self-efficacy, autonomy, and job impact. Together, these four elements enable employees to collaborate effectively, make decisions, and manage resources ([63]). Within the Job Demands-Resources (JD-R) model, the “attrition-path” hypothesis posits that excessive job demands can generate negative emotions when they exceed employees’ capacities. However, psychological empowerment, by interacting with job resources, can buffer the detrimental effects of demands on professional well-being ([5]; [64]). Leadership styles play a crucial role in this process. Research indicates that servant, transformational, and transactional leadership all significantly influence employees’ psychological empowerment ([60]), with servant leadership being particularly effective in strengthening teachers’ empowerment ([67]). Psychological empowerment, in turn, fosters positive emotions, self-efficacy, self-esteem, and self-confidence, all of which are closely linked to well-being ([25]). Empirical evidence also shows that psychological empowerment serves as a mediator in the relationship between leadership style and job satisfaction ([34]). Collectively, these findings highlight that organizational leaders can enhance employees’ professional well-being by addressing basic psychological needs and providing meaningful work-related autonomy. Therefore, the second hypothesis of this study is:

**H2.** 
*Servant leadership by principals significantly predicts kindergarten teachers’ professional well-being through the mediating role of teachers’ psychological empowerment.*


### 2.3. Inclusive Atmosphere as a Moderator

An inclusive climate integrates diverse personalities and perspectives, reduces bias, and fosters equality in organizational decision-making ([52]; [18]). As a form of organizational climate, it highlights leaders’ openness, fairness, and positivity, which in turn shape employees’ cognitions, attitudes, and behaviors ([3]). In diverse workplaces, an inclusive climate provides psychological safety, enabling employees to express themselves and fully utilize their talents ([48]). Empirical research shows that an inclusive climate enhances organizational commitment, promotes innovation, and positively moderates the relationship between servant leadership and employees’ occupational well-being ([17]). Within the Job Demands-Resources (JD-R) model, inclusive climates serve as job resources that motivate employees and support their career development ([5]). By ensuring respect and equality, they reduce interpersonal distance, build trust among colleagues, and foster positive manager–employee relationships ([52]; [47]). Hence, the third hypothesis of this study is:

**H3.** 
*An inclusive climate moderates the relationship between principals’ servant leadership and teachers’ psychological empowerment.*


In this context, we propose a moderated mediation model (Figure 1) that clearly illustrates how principals’ servant leadership influences teachers’ professional well-being.

## 3. Methods

### 3.1. Participants

This study used purposive sampling to investigate kindergarten teachers’ perceptions of principals’ servant leadership, the inclusive climate in kindergartens, teachers’ psychological empowerment, and their professional well-being. A total of 628 questionnaires were distributed in northeastern and central China, and 530 valid responses were collected, yielding a valid response rate of 84.39% after excluding questionnaires with unusually short completion times or inconsistent answers. Table 1 displays the details of the respondents.

### 3.2. Measures

In the survey, teachers provided their demographic information and their perceptions of kindergarten principal’s leadership, school climate, job satisfaction, and psychological empowerment. Data were analyzed using SPSS 26.0 and Process 3.3. First, SPSS 26.0 was used for data entry and organization, descriptive and correlation analysis. Afterwards, the SPSS macro program Process 3.3, developed by Hayes, was used to select the bootstrap method for estimating the confidence interval, with 5000 repetitions of sampling and the calculation of a 95% confidence interval.

#### 3.2.1. Kindergarten Teachers’ Professional Well-Being

This study adopted the Kindergarten Teacher Occupational Well-Being Scale (KTOWBS), revised by [71] ([71]). The scale consists of four dimensions: psychological well-being, emotional well-being, social well-being, and cognitive well-being, with a total of 15 items. The Cronbach’s alpha coefficients for each dimension range from 0.87 to 0.90. Emotional well-being refers to the fact that positive emotions prevail at work, with fewer negative emotions observed. Cognitive well-being refers to high satisfaction at work. Psychological well-being refers to the ability to control one’s work and have a sense of fulfillment. Social well-being refers to harmonious and good interpersonal relationships with one’s colleagues. Items 1 to 4 are for psychological well-being, 5 to 7 for emotional well-being, 8 to 11 for social well-being, and 12 to 15 for cognitive well-being. The answers to the questionnaire were based on a 3-point Likert scale, with scores ranging from a complete lack of conformity to complete conformity by 1~3 points. A total of 45 points could be obtained on the questionnaire, with higher scores indicating higher occupational well-being.

#### 3.2.2. Servant Leadership

This study employed the revised Servant Leadership Scale developed by [70] ([70]), as referenced by [76] ([76]). This scale integrates elements from earlier instruments developed by [40] ([40]), [54] ([54]), [16] ([16]), among other scholars. It comprises three dimensions: vision, service, and empowerment, with a total of 14 items. Specifically, items 1 to 4 correspond to the vision dimension, items 5 to 9 to the service dimension, and items 10 to 14 to the empowerment dimension. Responses were collected using a five-point Likert scale, ranging from 1 (strongly disagree) to 5 (strongly agree). The overall score was calculated as the mean of all items, with higher scores indicating a greater perceived level of servant leadership among kindergarten teachers. The scale demonstrated high reliability, with Cronbach’s alpha coefficients ranging from 0.91 to 0.94 across dimensions. To better align with the linguistic and contextual expressions of Chinese kindergarten teachers, minor modifications were made to certain items without altering the original scale structure. For example, “My supervisor often asks me about my opinion on the future direction of the company” was changed to “My leader often asks for my opinion on the future development of the kindergarten.” Similarly, “My supervisor prioritizes the personal development of departmental employees” was rephrased as “My leader prioritizes the development of kindergarten teachers.

#### 3.2.3. Psychological Empowerment

The Psychological Empowerment Scale was first developed by [62] ([62]) to better measure individual psychological empowerment, with its Cronbach’s alpha coefficients ranging from 0.79 to 0.85 for the dimensions of the source scale. [74] ([74]) used this scale in Chinese context, and its Cronbach’s alpha coefficient for each dimension ranged from 0.72 to 0.86 ([74]), indicating the suitability of this tool in China. The scale consists of four dimensions: job meaning, autonomy, self-efficacy, and job impact, with a total of 12 items. Items 1 to 3 are about job meaning, items 4 to 6 are about autonomy, items 7 to 9 are about self-efficacy, and items 10 to 12 are about job impact. The answers are based on a 5-point Likert scale. The higher the score of the servant leadership scale is, the greater the degree to which employees feel psychological empowerment. To make it more consistent with Chinese kindergarten context, this study modified expressions of some questions without changing the structure of the original scale. For example, “I have a powerful influence on the things that happen in my department” was modified to “I have a powerful influence on the teaching and learning that happens in the kindergarten.” And “I have a great deal of control over what happens in my department” was changed to “I have a great deal of control over what happens in kindergarten education”.

#### 3.2.4. Inclusive Climate

The Inclusive Climate Scale developed by [52] ([52]) was used. This scale includes three dimensions: hiring equity, the integration of differences, and the compatibility of decision-making, with a total of 15 items. Items 1 to 6 are about the integration of differences, items 7 to 10 are about the compatibility of decision-making, and items 11 to 15 are about the fairness of hiring. The answers to the questions were categorized on a scale of 1 to 7 using a 7-point Likert scale, with the smallest number representing complete disagreement ([52]). The Cronbach’s alpha coefficients of the scale range from 0.95 to 0.97, which is suitable for localized research in China. To ‘better align with the Chinese kindergarten context, this study adapted wording of certain questions without changing the structure of the original scale. For example, “The promotion procedure of the department is fair” was changed to “The promotion procedure of the kindergarten is fair”. And “The department is fair for all staff” was changed to “The promotion procedure of the kindergarten is fair”.

## 4. Results

### 4.1. Common Method Bias Test

The data in this study were obtained from self-reports, and common method bias may be present in the measurements. The study used “controlling for unmeasured single-method latent factor methods” to test for common method bias ([75]). First, a validated factor analysis model, Model F1, was constructed with the following leading fit indices: χ^2^/df = 4.277, NFI = 0.966, GFI = 0.924, CFI = 0.974, TLI = 0.967, and RMSEA = 0.079. Then, Model F2, containing method factors, was constructed based on the original validated factor analysis model, with the following main fit indices: χ^2^/df = 3.619, NFI = 0.977, GFI = 0.949, CFI = 0.983, TLI = 0.973, and RMSEA = 0.070. The difference between the fitted fit indices of Model F2 and Model F1 was slight, the indices of the CFI and TLI did not exceed 0.1, and the change in the RMSEA did not exceed 0.05. This indicated that the model was not significantly improved by adding the common method factor, and this study has no serious standard method bias.

### 4.2. Means, Standard Deviations, and Correlation Coefficients for Each Variable

Table 2 presents the mean, standard deviation, and correlation matrix of the variables. There is a significant correlation between kindergarten teachers’ professional well-being, principals’ servant leadership, and kindergarten teachers’ psychological empowerment. The results of the study provide data support for subsequent modeling.

### 4.3. Moderated Mediation Model Test

In the first step, a simple mediation model was tested. In the moderated mediation model, the simple mediation model is the benchmark model, which was tested first ([72]). In this study, Model 4 was first chosen to test the mediating role of psychological empowerment between principals’ servant leadership and kindergarten teachers’ professional well-being. The results of regression analysis showed that the demographic variables of kindergarten grade, kindergarten teachers’ age, education level, and enrollment status were significantly different for the variables of principals’ servant leadership, kindergarten teachers’ psychological empowerment, and kindergarten teachers’ occupational well-being, respectively. Thus, in controlling for kindergarten grade, as well as kindergarten teachers’ age, education level, and enrollment status, principals’ servant leadership significantly and positively predicted kindergarten teachers’ professional well-being (β = 0.269, *p* < 0.001). This proves that Hypothesis H1 is valid. When the mediator variable of kindergarten teachers’ psychological empowerment was introduced, principals’ servant leadership significantly and positively predicted kindergarten teachers’ psychological empowerment (β = 0.735, *p* < 0.001), and psychological empowerment was able to positively predict kindergarten teachers’ professional well-being (β = 0.137, *p* < 0.001). Moreover, the effect of principals’ ‘servant leadership on kindergarten teachers’ professional well-being remained significant (β = 0.101, *p* < 0.001). In sum, an equation model for the partial mediation role of psychological empowerment was established, with the partial mediation effect accounting for 49.070% of the total effect (Table 3). As such, Hypothesis H2 is supported.

### 4.4. Moderated Mediation Effects Test

In the second step, the mediating effect with moderation was tested. The model was chosen for testing since the moderating variables moderated the first half of the mediating effect. This study revealed that the demographic variables of kindergarten grade, kindergarten teachers’ age, education level, and enrollment status differed significantly for the variables of principals’ servant leadership, kindergarten teachers’ psychological empowerment, and kindergarten teachers’ professional well-being, respectively. Thus, in controlling for kindergarten grade as well as kindergarten teachers’ age, education level, title, and enrollment status, principals’ servant leadership positively predicted teachers’ psychological empowerment (β = 0.696, *p* < 0.001), and an inclusive climate positively predicted kindergarten teachers’ psychological empowerment (β = 0.145, *p* < 0.001). Moreover, the interaction term between principals’ servant leadership and an inclusive climate also significantly predicted teachers’ psychological empowerment (β = 0.074, *p* < 0.001), with a 95% confidence interval of [0.028, 0.120]. See Table 4 and Figure 2 for details.

To further understand the essence of the interaction between an inclusive climate and principals’ servant leadership, the inclusive climate was divided into a robust and inclusive climate group and a weak inclusive climate group according to one standard deviation of the mean. Table 5 depicts the mediation effect values and 95% bootstrap confidence intervals of teachers’ psychological empowerment between principals’ servant leadership and kindergarten teachers’ occupational well-being in the two groups of participants. Further simple slope analyses showed (e.g., Figure 3) that in the weakly inclusive climate group, principals’ servant leadership significantly and positively predicted kindergarten teachers’ psychological empowerment (βsimple = 0.696, t = 15.563, *p* < 0.001). In contrast, in the strongly inclusive climate group, principals’ servant leadership positively predicted kindergarten teachers’ psychological empowerment even more strongly (βsimple = 0.770, t = 7.029, *p* < 0.001). Taken together, these results indicate differences in the level of inclusive climate within different kindergartens, and that the predictive effect of principals’ servant leadership on kindergarten teachers’ psychological empowerment tends to increase as the overall level of the kindergarten’s inclusive climate increases. These results imply that Hypothesis H3 is valid.

## 5. Discussion

As societies advance, priorities have shifted from production efficiency to inner spirituality and experiential well-being, driving growing interest in more humanistic leadership models. Existing research on servant leadership has primarily examined its effects on employee job satisfaction ([39]), organizational citizenship behavior ([50]), and helping behavior ([69]) from a managerial perspective, while giving limited attention to teachers’ professional well-being in educational settings. This study addresses this gap by framing servant leadership as a key antecedent of kindergarten teachers’ professional well-being. It underscores the importance of leadership practices that enhance teachers’ job satisfaction and highlights the mechanisms through which servant leadership contributes to positive work experiences, workforce stability, and organizational efficiency. The findings provide practical insights for kindergarten administrators to refine leadership styles and adopt more humanistic management approaches that better support teachers’ professional well-being.

### 5.1. Servant Leadership and Kindergarten Teachers’ Professional Well-Being

Data analysis revealed a significant positive relationship between principals’ servant leadership and kindergarten teachers’ professional well-being, consistent with previous findings ([29]; [12]). Leadership style is a key determinant of employees’ mental health and occupational well-being ([32]), yet many studies have overlooked employee well-being as a primary outcome ([46]; [27]). Due to the demanding nature of their work, kindergarten teachers are particularly vulnerable to burnout and chronic stress, which can impair both their mental health and career development ([55]; [66]; [44]). Compared to other leadership styles, servant leadership places greater emphasis on teachers’ well-being by creating supportive work environments and addressing psychological needs ([53]; [30]). Guided by the Job Demands–Resources (JD-R) model, servant leadership provides teachers with critical resources and emotional support that mitigate job stress and enhance professional well-being ([15]). These findings highlight the importance for kindergarten principals to adopt servant leadership practices, foster a culture of respect and support, and actively promote teachers’ occupational well-being.

### 5.2. The Mediating Role of Psychological Empowerment

This study found that principals’ servant leadership enhances kindergarten teachers’ professional well-being by strengthening their psychological empowerment. Previous research has confirmed that psychological empowerment serves as a key mediating variable linking leadership style to employees’ perceptions and behaviors ([58]; [36]). According to the Job Demands–Resources (JD-R) model, servant leadership, grounded in service and collaboration, provides teachers with appropriate job resources, strengthens psychological empowerment, and promotes greater work engagement and job satisfaction ([5]).

In China, kindergarten teachers often face high parental expectations and heavy workloads ([8]), resulting in exhaustion and burnout ([26]) andhighlighting the need to improve their professional well-being. As a critical factor, psychological empowerment mediates the impact of servant leadership on teachers’ well-being. Therefore, kindergarten principals should recognize the multiple determinants of teachers’ psychological empowerment and adopt sustained strategies to strengthen both their empowerment and overall professional well-being.

### 5.3. The Moderating Role of an Inclusive Atmosphere

Data analysis indicates that an inclusive kindergarten climate moderates the relationship between principals’ servant leadership and teachers’ psychological empowerment, echoing findings from previous studies ([35]; [13]; [48]). Social context theory posits that team environment and organizational climate shape members’ perceptions and behaviors. A positive climate fosters satisfaction and creativity, whereas a hostile climate undermines well-being ([7]). Organizational climate can thus modulate the effects of leadership on employee attitudes and performance ([45]).

As a specific form of organizational climate, inclusiveness promotes equality, respect, and openness, encouraging employees to embrace diversity and address challenges collaboratively ([52]; [18]). In diverse work settings, an inclusive climate provides psychological safety, motivates talent utilization, and strengthens the employee–organization relationship ([17]; [10]; [41]). In this study, the moderating effect of an inclusive climate was stronger in kindergartens with higher inclusiveness, which enhanced teachers’ perception of positive leadership, their willingness to express themselves, and their sense of competence. These findings suggest that kindergarten principals should adopt servant leadership practices and foster an inclusive organizational climate to strengthen teachers’ psychological empowerment, organizational commitment, and overall professional well-being.

## 6. Limitations, Implications and Conclusions

This study examined the mechanism through which principals’ servant leadership affects kindergarten teachers’ professional well-being, but several limitations should be noted. First, due to constraints of time, geography, and researcher experience, the sample size was relatively small and unevenly distributed. In addition, purposive sampling may have introduced researcher bias, limiting the generalizability of the findings. Future studies may consider expand the sample size and diversity, covering different regions and participant groups to enhance representativeness. Second, the cross-sectional design limited the ability to examine temporal dynamics and establish causal relationships. Longitudinal designs with multiple data waves could better control extraneous factors and provide more precise insights. Third, the measures used, the Psychological Empowerment Scale and Inclusive Atmosphere Scale, were adapted from foreign instruments. Despite translation and validation efforts, cultural differences may have affected their construct validity and applicability in the Chinese preschool context. Fourth, reliance on self-reported data may have introduced social desirability bias, as teachers could overreport positive feelings or behaviors to align with professional or societal expectations, a limitation particularly salient in early childhood education research. Finally, the study focused only on the managerial perspective of how servant leadership influences teachers’ professional well-being. Future research should explore additional factors, introduce new variables, and further refine the underlying mechanisms. Despite these limitations, this study adopts the job demands-resources theory and social context theory as research perspectives, incorporating the variables of psychological empowerment and inclusive climate to explore the impact of servant leadership on the professional well-being of kindergarten teachers in depth, thereby further improving research in this field. It not only enriches the understanding of the internal mechanisms through which servant leadership affects the professional well-being of kindergarten teachers, but also further investigates the pathways through which servant leadership influences them, thus holding certain theoretical significance and practical value.

## Figures and Tables

**Figure 1 behavsci-15-01412-f001:**
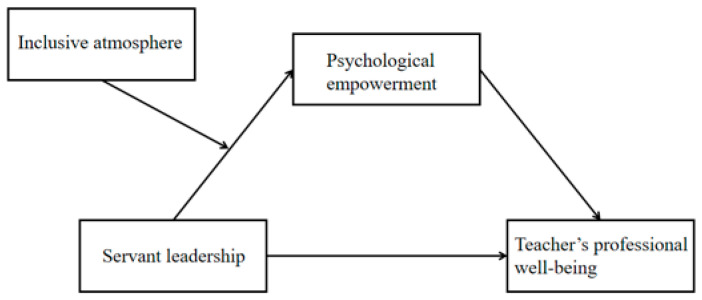
Moderated mediation model.

**Figure 2 behavsci-15-01412-f002:**
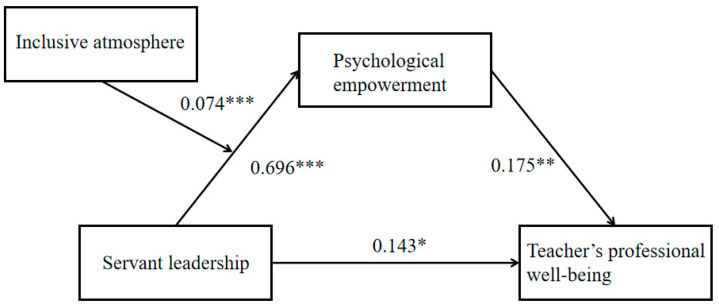
**Path diagram.** Note: *** *p* < 0.001; ** *p* < 0.01; * *p* < 0.05.

**Figure 3 behavsci-15-01412-f003:**
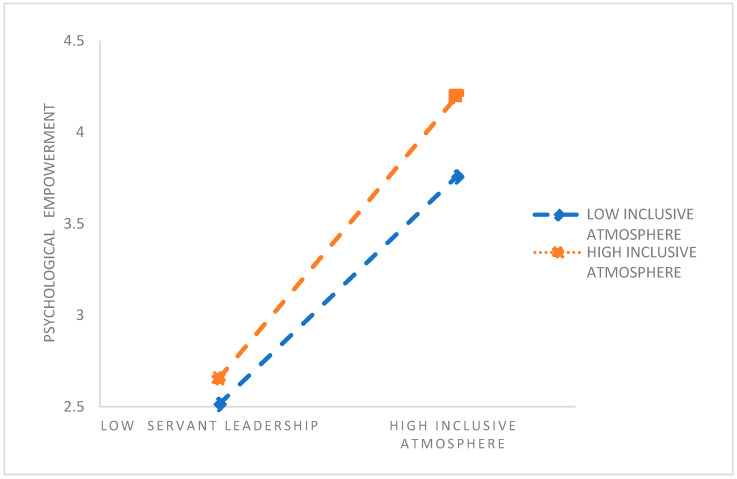
Slope plot.

**Table 1 behavsci-15-01412-t001:** List of basic information about the respondents to the questionnaire (*N* = 530).

	Form	Number	Percentage
1. Age	20 and under	6	1.10%
	21 to 30	304	57.30%
	31 to 40	137	26.00%
	41 and above	83	15.70%
2. Degree	high school or below	8	1.50%
	diploma	337	63.60%
	Bachelor’s degree	182	34.30%
	Mater’s degree or above	3	0.60%
3. Title	Unrated	287	54.20%
	Primary Title	148	27.90%
	Intermediate Title	63	11.90%
	Senior Title	32	6.00%
Kindergarten level	level III	116	21.90%
	level II	110	20.80%
	level I	304	57.40%
Establishment	Teachers in the establishment	201	37.90%
	Teachers outside the establishment	329	62.10%

Note: Kindergarten levels: Provincial level III refers to model kindergartens at provincial level. Level II refers to model kindergartens at city or county level. Level I refers to ordinary kindergartens.

**Table 2 behavsci-15-01412-t002:** Descriptive statistics for key variables and covariates (N = 530).

	M	SD	1	2	3	4
1. Kindergarten teachers’ professional well-being	1.863	0.473	1			
2. Servant Leadership	3.529	0.642	0.268 **	1		
3. Psychological empowerment	3.461	0.61	0.283 **	0.734 **	1	
4. An inclusive atmosphere	4.235	0.996	0.064	0.414 **	0.427 **	1

Note: ** *p* < 0.01.

**Table 3 behavsci-15-01412-t003:** Bootstrap analysis of mediation effects (*N* = 530).

	B	SE	LL 95% CI	UL 95% CI	Effect
Total effect	0.269	0.042	0.186	0.352	
Direct effect	0.137	0.062	0.016	0.258	50.93%
Indirect effect	0.132	0.044	0.048	0.221	49.07%

**Table 4 behavsci-15-01412-t004:** Mediated model tests with adjustment (n = 530).

Regression Equation (n = 530)		GFI		*p*		
Outcome	Predictor	*R*	*R* ^2^	*β*	*95% CI*	*t*
Psychological empowerment		0.756	0.571			
	Age			0.005	[−0.005, 0.015]	1.022
	Degree			−0.032	[−0.151, 0.086]	−0.539
	Title			0.026	[−0.080, 0.132]	0.485
	Establishment			−0.059	[−0.224, 0.105]	−0.710
	Kindergarten level			0.013	[−0.067, 0.093]	0.318
	Servant Leadership			0.696	[0.633, 0.760]	21.522 ***
	Inclusive atmosphere			0.145	[0.082, 0.207]	4.553 ***
	Service leadership × Inclusive atmosphere			0.074	[0.028, 0.120]	3.134 ***
kindergarten teachers						
Professional well-being						
	Age	0.324	0.105	−0.010	[−0.025, 0.004]	−1.409
	Degree			−0.151	[0.081, −0.322]	−1.748
	Title			0.131	[−0.021, 0.283]	1.688
	Establishment			0.112	[−0.124, 0.349]	0.935
	Kindergarten level			−0.083	[−0.198, 0.033]	−1.408
	Servant Leadership			0.143	[0.022, 0.264]	2.329 *
	Psychological empowerment			0.175	[0.055, 0.296]	2.852 **

Note: *** *p* < 0.001; ** *p* < 0.01; * *p* < 0.05.

**Table 5 behavsci-15-01412-t005:** The mediating effects of psychological empowerment.

Inclusive Atmosphere	*Effect*	*BootSE*	*BootLLCI*	*BootULCI*
*M* − 1*SD*	0.109	0.038	0.036	0.184
*M*	0.122	0.042	0.041	0.205
*M* + 1*SD*	0.135	0.047	0.045	0.228

## Data Availability

The original contributions presented in this study are included in the article. Further inquiries can be directed to the corresponding author.

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
