# Peer review of "The Influence of Servant Leadership on the Professional Well-Being of Kindergarten Teachers: A Moderated Mediation Model"

_behavsci, 2025, doi:10.3390/bs15101412_

Round 1
Reviewer 1 Report
Comments and Suggestions for Authors
I really like this study, it's a great much needed research topic. Servant leadership affects the professional well-being of kindergarten teachers in China. Teacher well-being is critical, as it reduces turnover, improves teaching quality, and supports healthier child development. With stress and burnout among kindergarten teachers becoming a serious issue, understanding the influence of leadership is essential.
I would improve the following:
In lines 59–66: The authors claim “few studies on preschool education” but later you cite several studies linking leadership styles to teacher well-being (Lines 90–109, 123–138). I think the novelty is overstated. You need to clarify what is new otherwise the contribution is incremental.
In lines 89–138: Hypotheses (H1–H2) simply restate well-established links. The paper reads more like replication than theory-building. I suggest pushing further and going deeper into how servant leadership is different than transformational leadership in educational contexts.
In lines 163–169: The purposive sampling is limited; as you mentioned briefly in the limitations section, it reduces generalizability. I think this weakness needs stronger acknowledgment.
In lines 186–253: the scales are adaptations of foreign instruments and the cultural translation process is not validated (e.g., back-translation, pilot testing). This raises the construct validity concerns.I think this also should be addressed more in the limitations.
In lines 254–267: The common method bias test is mentioned, but why not do the Harman’s single-factor or marker-variable methods. That would strengthen your claims.
In lines 268–295: The regression and mediation results are reported, however, the effect sizes and practical significance are underdeveloped.
In lines 367–421: it reads as a summary of results, not as a deep interpretation. I think you can engage with the contradictions, boundary conditions, and/or dive deeper into the cultural specificity.
In lines 422–449: the implications discussed are generic. There is no exploration of why inclusivity matters more in Chinese kindergartens. Perhaps mention something unique to Chinese cultural to fill this section with something more dynamic.
Lastly
In lines 452–479: the limitations are acknowledged but very standard.I think you can mention other factors such as self-report bias and social desirability in the field of childhood education
Author Response
Comments 1: In lines 59-66: The authors claim "few studies on preschool education" but later you cite several studies linking leadership styles to teacher well-being (Lines 90-109, 123-138). I think the novelty is overstated. You need to clarify what is new otherwise the contribution is incremental.
Response 1: Thank you for pointing this out. We have revised this section, elaborating on the necessity and innovativeness of studying teachers’ professional well-being from the perspective of the survival status of preschool teachers in China.
Comments 2: In lines 89–138: Hypotheses (H1-H2) simply restate well-established links. The paper reads more like replication than theory-building. I suggest pushing further and going deeper into how servant leadership is different than transformational leadership in educational contexts.
Response 2: Thank you for your suggestion. The mechanisms for forming the vision differ significantly between transformational and servant leadership: transformational leaders are oriented toward organizational interests, while servant leaders are committed to enhancing member well-being. Previous studies have also indicated differences in their approaches to building trust—servant leaders establish trust through emotional bonds, whereas transformational leaders emphasize cognitive-level trust. We have restructured and refined this section, moving beyond a simple description of the association between servant leadership and preschool teachers’ well-being, and instead integrating theoretical foundations, empirical evidence, and the distinctive features of servant leadership.
Comments 3: In lines 163-169: The purposive sampling is limited; as you mentioned briefly in the limitations section, it reduces generalizability. I think this weakness needs stronger acknowledgment.
Response 3:Thank you for your professional advice. We acknowledge that purposive sampling may introduce researcher bias, which limits the generalizability of the conclusions. Future studies should expand the sample size and diversity, covering different regions and participant groups to enhance representativeness. This limitation has been added to the "Limitations" section.
Comments 4: In lines 186-253: the scales are adaptations of foreign instruments and the cultural translation process is not validated (e.g., back-translation, pilot testing). This raises the construct validity concerns. I think this also should be addressed more in the limitations.
Response 4: Thank you for your valuable comment. The psychological empowerment scale and inclusive climate scale used in this study were adapted from instruments developed abroad. Although they have been translated and validated, cultural differences may affect their structural validity and applicability in the Chinese preschool education context. This limitation has been addressed in the revised manuscript.
Comments 5: In lines 254-267: The common method bias test is mentioned, but why not do the Harman's single-factor or marker-variable methods. That would strengthen your claims.
Response 5: The latent factor method, through confirmatory factor analysis, allows for constructing more complex models (e.g., models with and without a common method factor) and comparing chi-square differences between them. This approach provides a more comprehensive examination of variable relationships and a more accurate assessment of whether common method bias exists. Therefore, this method was primarily adopted in the present study to ensure the validity of the results.
Comments 6: In lines 268-295: The regression and mediation results are reported, however, the effect sizes and practical significance are underdeveloped.
Response 6: Thank you for your professional advice. This study highlights the importance of leadership practices in improving teachers' job satisfaction and demonstrates how servant leadership functions by enhancing work experiences, strengthening team cohesion, and optimizing organizational effectiveness. These practical implications have been further elaborated in the discussion section.
Comments 7: In lines 367-421: it reads as a summary of results, not as a deep interpretation. I think you can engage with the contradictions, boundary conditions, and/or dive deeper into the cultural specificity.
Response 7: Thank you for your insightful comment. In response, we have expanded the discussion section with a more in-depth analysis of the issue you raised.
Comments 8: In lines 422-449: the implications discussed are generic. There is no exploration of why inclusivity matters more in Chinese kindergartens. Perhaps mention something unique to Chinese culture to fill this section with something more dynamic.
Response 8: Thank you for your professional suggestion. Indeed, the current preschool teacher workforce in China faces challenges such as heavy work pressure, relatively low overall quality, high turnover, and strong intentions to leave. Their professional survival environment deserves further improvement. The inclusion of the variable "inclusive climate" provides a new pathway to explore the enhancement of teachers' professional well-being at the organizational climate level. This content has been added to the main text.
Comments 9: In lines 452-479: the limitations are acknowledged but very standard. I think you can mention other factors such as self-report bias and social desirability in the field of childhood education
Response 9: Thank you for highlighting this important point. Since this study relies on self-reported data from kindergarten teachers, social desirability bias may have affected responses. Teachers may have exaggerated positive emotions or behaviors to align with professional or societal expectations. This limitation has been addressed in the revised manuscript.
Reviewer 2 Report
Comments and Suggestions for Authors
This manuscript examines the relationships among servant leadership, the professional well-being of kindergarten teachers, psychological empowerment, and an inclusive atmosphere. With a large sample (n = 530), validated measurement instruments, and a structured statistical model, the study offers a potentially useful contribution to the literature on leadership and teacher well-being. The paper is generally well-structured; however, several important issues limit its contribution in the current form. In particular, the link to the JD–R theoretical framework is underdeveloped, the novelty of the study remains unclear, and several sections of the manuscript would benefit from clearer organization and more concise writing.
Strengths:
• Sample size: The study employs a substantial sample of 530 kindergarten instructors, augmenting statistical power and the generalizability of results.
• Measurement: Established, dependable, and valid scales were utilized.
• Design: The authors used a statistical model to show how variables are related to each other, and they also did a test for common method bias.
Major Comments :
- Theoretical Framing and Contribution
- The results show that servant leadership, empowerment, and inclusive climate all contribute positively to teachers’ well-being. While interesting, these results are not surprising, as most resource-oriented variables tend to correlate positively with well-being. The manuscript must articulate its novelty and contribution more explicitly. What does this study add that was not known before? Is it the focus on kindergarten teachers, the specific mediating mechanism, or the inclusion of inclusive climate? A clear “Contribution” paragraph at the end of the Introduction or Discussion would strengthen the paper.
- JD–R Framework
- The authors claim that the study is based on the JD–R model. However, no job demand variable is included. This is problematic because JD–R fundamentally examines the balance between job demands and job resources.
- The context of kindergarten teaching naturally involves significant demands (physical fatigue, constant vigilance, parental pressure). At least one such demand should have been included in the model to justify using JD–R.
- If integrating demands is not possible with the current dataset, then the authors should either (a) remove JD–R as the central framework and use an alternative theoretical base (e.g., COR theory, social exchange theory), or (b) acknowledge this limitation explicitly and adjust their theoretical claims.
- Clarity of Constructs (Perceived Leadership)
- The study does not measure leaders’ servant leadership directly but rather teachers’ perceptions of their leaders’ servant leadership. This distinction should be explicitly acknowledged in the abstract, introduction, and methodology.
- Organization of the Paper
- The model figure should appear at the end of the Introduction, to help readers clearly understand the hypothesized relationships.
- Hypotheses should be presented in a distinct and numbered format, not embedded within long paragraphs. This will improve readability.
- The Results section is somewhat difficult to follow. It should be reorganized to first present descriptives and correlations, then the measurement model, and finally the structural model and hypothesis testing. Use tables to display key coefficients and fit indices.
- The Introduction currently reads more like a theoretical overview rather than a problem-driven opening. Please include a clear statement of the research problem, why it matters in the kindergarten context, and how the study addresses a gap.
- Much of the Discussion repeats results in narrative form. Instead, focus on:
- Theoretical implications: How do these findings extend or challenge existing leadership and well-being theories?
- Practical implications: What concrete actions can school leaders and policymakers take based on these results?
- Subsections like “Mediation” and “Moderation” are not necessary in the Discussion unless they are tied to broader theoretical insights.
- Limitations and Future Research :Currently, these sections are blended together. Limitations include: cross-sectional design, single-source perceptual data, and lack of job demands in the JD–R model.
- Language and Style :The writing is sometimes complex and difficult to follow. Shorter sentences and clearer paragraph breaks would improve readability. A professional language edit is strongly recommended.
Overall, this is a potentially valuable study. However, the theoretical framing, contribution, and clarity of presentation need significant improvement. The manuscript would benefit most from (a) clarifying its theoretical framework (JD–R vs. alternatives), (b) articulating its novelty, (c) reorganizing key sections, and (d) improving the writing style.
Author Response
Comments 1: The manuscript must articulate its novelty and contribution more explicitly. What does this study add that was not known before? Is it the focus on kindergarten teachers, the specific mediating mechanism, or the inclusion of inclusive climate? A clear "Contribution" paragraph at the end of the Introduction or Discussion would strengthen the paper.
Response 1: Thank you for your constructive suggestions. We have revised both the introduction and discussion sections to elaborate on the innovativeness of this study from three perspectives: the survival status of preschool teachers in China, the innovativeness of the research population, and the innovativeness of the research mechanism.
Comments 2: If integrating demands is not possible with the current dataset, then the authors should either (a) remove JD–R as the central framework and use an alternative theoretical base (e.g., COR theory, social exchange theory), or (b) acknowledge this limitation explicitly and adjust their theoretical claims.
Response 2: Thank you for your valuable suggestion. This study primarily emphasized the gain-path hypothesis of the JD-R model without including job demand variables, which makes it less suitable to present it as the central theoretical model. However, the gain-path hypothesis does state that when leaders, supervisors, or colleagues provide employees with support, trust, and assistance, employees can develop occupational well-being. In this study, servant leadership provides preschool teachers with job resources that, in interaction with psychological empowerment, effectively mitigate the negative effects of job demands and enhance professional well-being. Accordingly, we have adjusted the framework of the paper, presenting the JD-R model, together with social exchange theory, as theoretical underpinnings rather than the sole theoretical model.
Comments 3: The study does not measure leaders' servant leadership directly but rather teachers' perceptions of their leaders' servant leadership. This distinction should be explicitly acknowledged in the abstract, introduction, and methodology.
Response 3: Thank you for your professional suggestion. There are two reasons why this study evaluated principals' servant leadership behavior from teachers' perspectives. First, assessing principals' leadership behavior from the teachers' perspective avoids the influence of principals' subjective emotions, making the results more objective. Second, many existing studies have also assessed principals' servant leadership behavior from teachers' perspectives, and this method has empirical support. Nevertheless, we value your suggestion. Since this study relies on self-reported data from kindergarten teachers, responses may be influenced by social desirability bias, with teachers exaggerating positive emotions or behaviors to meet professional or social expectations. This limitation has been noted in the revised manuscript.
Comments 4: The model figure should appear at the end of the Introduction, to help readers clearly understand the hypothesized relationships.
Response 4: Thank you for your suggestion. We have adjusted the placement of the model diagram accordingly.
Comments 5: Hypotheses should be presented in a distinct and numbered format, not embedded within long paragraphs. This will improve readability.
Response 5: Thank you for your suggestion. Thank you for your suggestion. We have now listed the research hypotheses separately.
Comments 6: The Results section is somewhat difficult to follow. It should be reorganized to first present descriptive statistics and correlations, then the measurement model, and finally the structural model and hypothesis testing. Use tables to display key coefficients and fit indices.
Response 6: Thank you for your suggestion. Based on your advice and that of other scholars, we have partially revised the structure of the results section.
Comments 7: The Introduction currently reads more like a theoretical overview rather than a problem-driven opening. Please include a clear statement of the research problem, why it matters in the kindergarten context, and how the study addresses a gap.
Response 7: Thank you for your constructive suggestion. In the introduction, we have elaborated on the necessity of studying preschool teachers' well-being in the kindergarten context, and further highlighted the innovativeness of this study in terms of the survival status of preschool teachers in China, the research population, and the research mechanism.
Comments 8: Much of the discussion repeats results in narrative form. Instead, focus on: Theoretical implications: How do these findings extend or challenge existing leadership and well-being theories? Practical implications: What concrete actions can school leaders and policymakers take based on these results?
Response 8: Thank you for your valuable comment. This study emphasizes the importance of leadership practices in enhancing teachers’ job satisfaction and illustrates how servant leadership improves work experiences, strengthens team cohesion, and enhances organizational effectiveness. These practical implications have been further explained in the discussion section.
Comments 9: Limitations and Future Research :Currently, these sections are blended together. Limitations include: cross-sectional design, single-source perceptual data, and lack of job demands in the JD–R model.
Response 9: Thank you for your professional advice. The limitations you mentioned, such as the cross-sectional design and reliance on single-source data, have been addressed in the revised manuscript. We have also refined the language to make the writing more concise and coherent.
Reviewer 3 Report
Comments and Suggestions for Authors
The influence of servant leadership on the professional well-being of kindergarten teachers: A moderated mediation model
Thanks to the editor for allowing me to read and review this exciting topic. I appreciate the effort the authors put into writing this research article.
- The abstract is convincing. Add the methodology used in the study.
- The introduction needs improvement; it could be enhanced to provide the reader with a clear understanding of the study. The introduction should objectively present the topic and the main concepts and literature, explaining the relevance and topicality of the subject (suggestion: 1 paragraph). Based on that, it should describe a research gap, problem, and aim (suggestion: 2 or 3 paragraphs, with 1 highlighting the objective). The idea is to emphasize the research gap, providing stronger arguments about its relevance, impact, and innovation. Please present the problem statement and research objectives.
- The theoretical framework and literature review are convincing. Add a conceptual framework figure before the methods.
- The results section is convincing.
- The discussion section is convincing.
- Make a separate section of conclusion with theoretical implications and managerial contributions.
- Add limitations and future research after the conclusion and implications
- References need to be updated to the recent studies. Also, to support the writing, include more studies from 2022, 2023, 2024, and 2025.
All the Best
Author Response
Comments 1: The introduction needs improvement; it could be enhanced to provide the reader with a clear understanding of the study. The introduction should objectively present the topic and the main concepts and literature, explaining the relevance and topicality of the subject (suggestion: 1 paragraph). Based on that, it should describe a research gap, problem, and aim (suggestion: 2 or 3 paragraphs, with 1 highlighting the objective). The idea is to emphasize the research gap, providing stronger arguments about its relevance, impact, and innovation. Please present the problem statement and research objectives.
Response 1: Thank you for your constructive suggestions. We have revised the introduction and discussion sections to elaborate on both the innovativeness and the practical implications of this study, particularly in terms of the survival status of preschool teachers in China, the innovativeness of the research population, and the innovativeness of the research mechanism. The research objectives have also been further clarified.
Comments 2: The theoretical framework and literature review are convincing. Add a conceptual framework figure before the methods.
Response 2: Thank you for your suggestion. We have added a conceptual framework figure before the methods.
Comments 3: Make a separate section of conclusion with theoretical implications and managerial contributions.
Response 3: In terms of practical significance, this study emphasizes the importance of leadership practices in enhancing teachers' job satisfaction and illustrates how servant leadership improves work experiences, strengthens team cohesion, and enhances organizational effectiveness. These practical implications have been further explained in the discussion section.
Comments 4: Add limitations and future research after the conclusion and implications.
Response 4: Regarding the limitations of this study and future research directions, we acknowledge that purposive sampling may introduce researcher bias, which limits the generalizability of the conclusions. Future studies should expand the sample size and diversity, covering different regions and participant groups to enhance representativeness. Additionally, the psychological empowerment scale and inclusive climate scale used in this study were adapted from instruments developed abroad. Although they have been translated and validated, cultural differences may affect their structural validity and applicability in the Chinese preschool education context. These limitations have been added to the "Limitations" section.
Round 2
Reviewer 2 Report
Comments and Suggestions for Authors
I would like to thank the authors for their careful consideration of the previous comments and for the substantial improvements made in the revised version of the manuscript.
- Novelty and Contribution: The authors have now explained the novelty and contribution of the article more clearly, particularly in three respects: the survival status of preschool teachers in China, the innovativeness of the research population, and the innovativeness of the research mechanism.
- Theoretical Framing: The perspective related to the JD-R model has been elaborated in a more explanatory manner, which strengthens the theoretical foundation.
- Measurement Issues: The emphasis on perceived servant leadership has been clarified, and earlier misunderstandings have been resolved.
- Presentation of Model and Hypotheses: The presentation of the model and hypotheses has been made clearer and easier to follow.
- Introduction: The introduction has been re-elaborated in line with the Chinese context, which improves the coherence of the paper.
- Discussion and Limitations: The discussion, limitations, and future research sections are now better structured and more aligned with the findings.
Overall, the paper is now well written and much clearer than in the previous version. While the manuscript is more polished, it should be noted that the scientific contribution is moderate rather than strong.
Comments on the Quality of English LanguageIt would be beneficial for the text to be reviewed one last time by a native English speaker.
Author Response
Comments 1: While the manuscript is more polished, it should be noted that the scientific contribution is moderate rather than strong.
Response 1: Thank you for pointing this out. Based on your suggestions, we have further elaborated on the theoretical value and practical significance of this research in the section on limitations and future directions. This study examines the impact of kindergarten principals' leadership on teachers' professional well-being and delves into the underlying mechanisms. By introducing the variables of psychological empowerment and inclusive atmosphere, it not only enriches the theoretical framework surrounding teacher well-being but also offers new practical insights for educational practitioners. Therefore, this research warrants further attention within the field of early childhood education.
Comments 2: It would be beneficial for the text to be reviewed one last time by a native English speaker.
Response 2: Thank you for your suggestion.Based on your suggestions, we have further refined the language in the paper to better align with English expression conventions and academic requirements for publication.